# Improving the Activity of Tryptophan Synthetase via a Nucleic Acid Scaffold

**DOI:** 10.3390/molecules28217272

**Published:** 2023-10-26

**Authors:** Yaping Wang, Xiangyi Wang, Shuhui Niu, Wei Cheng, Xiaoyan Liu, Yong Min, Yimin Qiu, Lixin Ma, Ben Rao, Lei Zhu

**Affiliations:** 1Hubei Biopesticide Engineering Research Center, Hubei Academy of Agricultural Sciences, Biopesticide Branch of Hubei Innovation Centre of Agricultural Science and Technology, Wuhan 430064, China; 2State Key Laboratory of Biocatalysis and Enzyme, Engineering Hubei Collaborative Innovation Center for Green Transformation of Bio-Resources, Hubei Key Laboratory of Industrial Biotechnology, Biology Faculty of Hubei University, Hubei University, Wuhan 430062, China

**Keywords:** tryptophan synthetase, dCE, TrapA, TrapB, nucleic acid scaffold

## Abstract

Tryptophan synthetase (TSase), which functions as a tetramer, is a typical enzyme with a substrate channel effect, and shows excellent performance in the production of non-standard amino acids, histamine, and other biological derivatives. Based on previous work, we fused a mutant CE protein (colistin of *E. coli*, a polypeptide with antibacterial activity) sequence with the sequence of TSase to explore whether its catalytic activity could be enhanced, and we also analyzed whether the addition of a DNA scaffold was a feasible strategy. Here, dCE (CE protein without DNase activity) protein tags were constructed and fused to the TrapA and TrapB subunits of TSase, and the whole cell was used for the catalytic reaction. The results showed that after the dCE protein tag was fused to the TrapB subunit, its whole cell catalytic activity increased by 50%. Next, the two subunits were expressed separately, and the proteins were bound in vitro to ensure equimolar combination between the two subunits. After the dCE label was fused to TrapB, the activity of TSase assembled with TrapA also improved. A series of experiments revealed that the enzyme fused with dCE9 showed higher activity than the wild-type protein. In general, the activity of assembly TSase was optimal when the temperature was 50 °C and the pH was about 9.0. After a long temperature treatment, the enzyme maintained good activity. With the addition of exogenous nucleic acid, the activity of the enzyme increased. The maximum yield was 0.58 g/L, which was almost three times that of the wild-type TSase (0.21 g/L). The recombinant TSase constructed in this study with dCE fusion had the advantages of higher heat resistance and higher activity, and confirmed the feasibility of adding a nucleic acid scaffold, providing a new idea for the improvement of structurally similar enzymes.

## 1. Introduction

Tryptophan synthetase (TSase) is an enzyme that consists of subunits α (TrapA) and β (TrapB). The subunits can catalyze the formation of L-tryptophan. Usually, two α subunits wrap around two β subunits to form a tetramer; they can catalyze chemical reactions alone, as well as cooperate with each other to enhance their catalytic activity [1,2,3].

Due to its unique structure, TSase catalyzes chemical reactions in two steps (Figure 1). The α subunit catalyzes the anti-hydroxylic reaction of indole-3-glycerophosphate (IGP) to generate glyceraldehyde-3-phosphate (GAP) and indole. Next, indole reaches the active center of the β subunit, and indole and L-serine are converted to L-tryptophan under the catalytic action of pyridoxal phosphate (PLP). When serine reacts with PLP, lysine residues of the β subunits combine to form the internal aldehyde group (IA), which is also an important intermediate in the biosynthesis of amino acrylate (AA). Then, indole and the active site of the β subunit react with the generated AA, forming tryptophan and regenerating IA, ensuring that the catalytic reaction can continue [3,4,5,6,7,8,9,10,11,12].

In industrial engineering, the use of enzymes is increasing due to their environmental friendliness. TSase, as an important enzyme with excellent thermal stability, high reaction activity, and wide substrate adaptability, is widely used in various fields. TSase can catalyze not only the synthesis of L-tryptophan, but also the production of more than 30 different non-standard amino acids when indole or indole analogs are used [8,13,14,15,16].

In nature, in order to maintain normal cellular physiological activities, a variety of enzymes need to cooperate with each other. Inside cells, enzymes of various cascade reactions can form multi-enzyme complexes through combination, connecting the active sites and promoting the formation of substrate channels. However, engineering of such systems is not easy due to various reasons, such as inefficient use of substrates, unstable and/or toxic intermediates, or undesirable side effects. In-depth research on multi-enzyme reactions revealed that most reactions maximize the performance of multi-enzyme complexes by placing the enzymes as close together as possible. Therefore, nucleic acids and proteins have been used as scaffolds to shorten the distance between enzymes and accelerate the reaction [17,18,19].

Inside cells, many complex materials are formed by combining DNA and protein, which can be used to generate DNA scaffolds. The number of proteins bound to the DNA scaffold can be controlled by artificially increasing or decreasing the number of specific sequence repeats; DNA also has the advantage in that the structure of some key sites can be accurately predicted, so the sequence of nucleotides can be changed according to the preference of the binding protein, thus changing the protein binding properties to the scaffold. Since DNA has its own double helix structure, its stability is not dependent on the length of the nucleotide sequence, so the number of bound proteins can be freely increased. Colistin is a polypeptide with antibacterial activity that is produced by Gram-positive and Gram-negative bacteria. CE is a common colistin, which has a strong ability to bind to DNA, as well as a large number of nucleic acids inside cells, providing a basis for the use of nucleic acid scaffolds in engineering bacteria [20,21,22,23].

Here we report an exploratory experiment based on the previous work of our laboratory, fusing modified CE and TSase. Modified CE no longer exhibited DNase activity, but retained its ability to bind to nucleic acids. We explored whether the introduction of the dCE protein as a nucleic acid scaffold can improve TSase activity. The output of the target product was indeed increased, providing a reference for the modification of similar enzymes and the application of nucleic acid scaffolds in future studies.

## 2. Results and Discussion

### 2.1. The Expression of Fusion Proteins Ectrp-dCEs

TSase is composed of subunits α and β forming an αββα tetramer structure, so the gene fragment of TSase is also composed of the *TrapA* gene and the *TrapB* gene placed in tandem. First, the *TrapA* gene N end was fused with dCE tags to obtain a series of recombinant plasmids that can express Ectrp-dCE2, Ectrp-dCE7, Ectrp-dCE8, and Ectrp-dCE9, respectively. Protein expression was induced. The cells were disrupted, and the protein was purified. The obtained solutions were subjected to SDS-PAGE for detection.

Here, the SDS-PAGE (Figure 2) showed the Ectrp-dCE2 expression. Apparently, the dCE2 label was fused to the α subunits. The size of the TrapA protein with label was about 44 kDa, and the size of the TrapB protein without label was about 47 kDa. Both bands could be clearly seen in the figure. It is obvious that Ectrp-dCE2 was successfully expressed by using *E. coli.* Results showed that Ectrp-dCE7, Ectrp-dCE8, and Ectrp-dCE9 were also successfully expressed, respectively.

The recombinant strains containing these expression plasmids (Ectrp-dCE2, Ectrp-dCE7, Ectrp-dCE8, and Ectrp-dCE9) were used to synthesize L-tryptophan. L-tryptophan levels were detected by HPLC, and the results are shown in Figure 3; they show that there was no difference between the activity of the labeled and non-labeled TSase. Hence, the label was subsequently fused to the β subunit.

### 2.2. The Expression of Fusion Proteins dCEs-Ectrp

Next, we fused dCE tags to the β subunit to obtain dCE2-Ectrp, dCE7-Ectrp, dCE8-Ectrp, and dCE9-Ectrp. *E. coli* BL21 was transformed, and protein expression was induced. The cells were disrupted, and the protein was purified. SDS-PAGE was applied to detect the protein’s expression.

Here, the SDS-PAGE (Figure 4) showed the dCE2-Ectrp expression. The size of the TrapB protein fused with dCE labels was about 64 kDa, and the size of the TrapA protein without a label was about 27 kDa, which are clearly shown in Figure 3. In addition, a band at 47 kDa was observed below all dCE-labeled TrapB samples, possibly indicating the presence of an un-labeled TrapB protein. The linker may break during translation and folding of the expressed protein, causing the label fail to fuse, and changing the ratio between the two subunits. It is obvious that dCE2-Ectrp was successfully expressed by using *E. coli.* Results showed that dCE7-Ectrp, dCE8-Ectrp, and dCE9-Ectrp were also successfully expressed, respectively.

The recombinant strains containing these expression plasmids (dCE2-Ectrp, dCE7-Ectrp, dCE8-Ectrp, and dCE9-Ectrp) were used to synthesize L-tryptophan. L-tryptophan levels were detected by HPLC, and results are shown in Figure 5. It is obvious that the labeled enzymes produced more L-tryptophan than the un-labeled TSase. Hence, it was preferable to fuse the β subunit to improve the TSase activity. Originally, one plasmid was used to express both subunits. Next, we chose to express the two subunits separately, and the proteins were combined in vitro to ensure the equimolar combination between the two subunits and improve the activity of labeled TSase.

### 2.3. Separate Expression of TrapA and dCE-Labeled TrapB

TSase consists of two subunits forming a tetramer, so it is very important to ensure equal proportions of binding between the two proteins. Based on the differences in effective expression between labeled α and β subunits, we decided to construct a pET23a TrapB-dCE vector series and pET23a TrapA, and used pure enzymes for subsequent determination of enzyme activity and other aspects.

When selecting the carrier to construct the dCE-labeled TrapB proteins, the labels were fused to the TrapB C-terminus. The positive colonies were selected, and plasmids were sequenced by an external company. Sequencing results were correct. The plasmid was transformed into the expression vector and then purified (Figure 6).

The recombinant strains expressing four newly constructed fusion proteins were cultured and the protein was purified. The eluate with the highest purity was selected for ultrafiltration concentration. These TrapB with a C-terminally fused dCE tag had a molecular weight of about 64 kDa. TrapA was also expressed and purified.

### 2.4. In Vitro Assembly of Tryptophan Synthetase

Here, the TrapB-dCE8 fusion protein was chosen to perform the assembly experiment with TrapA. The TrapB-dCE8 fusion protein and TrapA were purified and concentrated. The two proteins were incubated in vitro at a low temperature for 30 min, and then gel filtration was conducted on the protein mixture to detect whether the two proteins could be successfully combined into a proportionally correct and complete TSase in vitro.

It can be seen from the molecular sieve results (Figure 7) that there were three peaks. In order to detect the protein components contained in each peak, each peak was sampled, and SDS-PAGE (Figure 8) was performed.

Obviously, in the first peak, TrapA and TrapB proteins were assembled in vitro and bound together by the interaction between proteins. In the second peak, excessive TrapA protein was present. The third peak contained no protein. This proved that the two TSase subunits could be correctly assembled into TSase in vitro, laying a foundation for subsequent research. The assembly experiments of other dCE-labeled TrapB proteins with TrapA were performed, and results showed that they could all be assembled into TSase in vitro.

### 2.5. Reaction and Detection of Assembled-TSase

The various TrapB-dCE and TrapA proteins were mixed at an equal molar ratio and incubated, and then the reaction was started. The products were analyzed using thin layer chromatography (Figure 9) and HPLC (Figure 10).

The thin layer chromatography results showed that assembled TSase could catalyze the synthesis of tryptophan in vitro. To further analyze the effects of dCE labels on the activity of TSase, quantitative HPLC analysis was conducted. After the dCE tag was fused at the C-terminus of TrapB, L-tryptophan could be normally generated, and the enzyme activity improved significantly. The dCE9 fused TrapB with TrapA showed the highest enzyme activity.

### 2.6. Characterization of Assembled Tryptophan Synthetase

Since dCE proteins can bind to nucleic acids, we treated dCE-labeled TrapB proteins to remove nucleic acid using a heparin column.

The enzyme activity was tested in the temperature range of 25–60 °C in steps of 5 °C to determine the optimal reaction temperature (Figure 11). The enzymatic reactions were performed, and HPLC was performed for quantitative analysis of L-tryptophan. The tryptophan synthetase activity was calculated. The activities of all assembled TSases were normalized to the highest activity of non-labeled TSase (Ectrp).

The optimal reaction temperature was found to be 40 °C. At 60 °C, we still observed catalytic activity, proving that TSase was still active at high temperatures. After labeling with dCEs, the activity of the enzyme was inhibited at low temperature, but there was an upward trend at 30 °C, and the catalytic activity was higher than at 35 °C. A temperature of 30 °C may be more conducive to the conformational change of the recombinant TSase, accelerating the reaction. With increasing temperatures, the activity of recombinant TSase further increased, reaching the highest value at 50 °C. At higher temperatures, the enzyme activity gradually decreased, but remained higher than that of unlabeled TSase. The assembled TSase was again purified by a heparin column. After the endogenous nucleic acid was removed, the treated recombinant TSase was treated with a temperature gradient in the same way. Obviously, the activity of the recombinant TSase was significantly decreased compared with the enzyme without heparin column purification.

The pH value affects enzyme activity mainly by changing the conformation of the active site of the enzyme (Figure 12). Therefore, we analyzed the activity of assembled TSase at various pH values.

Unlabeled TSase had the highest activity at pH 9.0. The optimal pH of the assembled dCE-labeled recombinant TSase was also around 9.0. In different pH environments, the activity of the labeled TSase was higher than that of the non-labeled TSase.

After the assembled TSase was purified with a heparin column to remove endogenous nucleic acid, the reaction was carried out under the same conditions. The overall activity of labeled enzymes was slightly higher than that of non-labeled TSase. However, compared with the labeled TSase containing endogenous nucleic acids, TSase activity was significantly decreased. The optimum pH was still around 9.0.

Next, we tested the thermal stability (Figure 13). Tris-HCl buffer at pH 9.0 was used to prepare the reaction solution, and labeled and non-labeled TSase were incubated at 50 °C for 0, 4, 8, and 12 h. The reaction was carried out and the enzyme activity was determined.

Non-labeled TSase showed increased activity after 4 h of incubation. The temperature treatment may change the structure of the enzyme, accelerating the reaction. The enzyme activity was the highest after 8 h of heat treatment, and decreased thereafter. It may be that after fusion of the dCE label, the nucleic acid combines with the enzyme, so after a long temperature treatment, the conformation of the enzyme and nucleic acid has changed, accelerating the enzymatic reaction. Moreover, it can be seen that the activity of assembled TrapB-dCE8/TrapA, TrapB-dCE9/TrapA are much higher than that of TSase without the dCE tag, both with and without temperature treatment.

However, after the assembled enzyme was purified by heparin column to remove endogenous nucleic acid, it was treated at a temperature of 50 °C for different periods. The results show that after temperature treatment, the enzyme activity after the removal of endogenous nucleic acid was much lower than that of the enzyme with nucleic acid. After the removal of nucleic acid, after a long period of temperature treatment, only the enzyme activity of TrapB-dCE9 was still higher than that of non-labeled enzyme. The optimal temperature treatment time for removing endogenous nucleic acid was about 8 h.

### 2.7. Using Exogenous Nucleic Acids to Improve the Enzyme Activity of Fusion Protein

The activity of assembled TSase was significantly increased than un-labeled TSase. After the removal of endogenous nucleic acids, the activity of the assembled enzyme was significantly decreased. Therefore, we aimed to increase the activity of assembled TSase by adding exogenous nucleic acid.

After TrapB-dCE proteins were purified by heparin column to remove the bound endogenous nucleic acid, exogenous nucleic acid substances were added at different ratios, followed by incubation at 37 °C for 1 h, and then nucleic acid dye was added in proportion for a DNA mobility shift assay (Figure 14) to observe whether it can combine with nucleic acid.

Wild-type TSase does not have the ability to bind with nucleic acid (Figure 15a). The TrapB-dCE proteins purified by heparin could bind exogenous nucleic acids (Figure 15b), and the binding ability was roughly the same, basically starting when the molar ratio of protein to exogenous nucleic acid was about 2500 (Figure 16). Higher amounts of exogenous nucleic acid resulted in increased binding. Here, nucleic acids and dCE-labeled proteins have been used as scaffolds to shorten the distance between enzymes and accelerate the reaction.

The enzyme activities at different exogenous nucleic acid concentrations were normalized to the activity of a recombinant enzyme without nucleic acid and plotted. After the removal of endogenous nucleic acid and the addition of exogenous nucleic acid, the activity of all assembled TSases improved, among which the activity of the dCE9-labeled protein was the most affected. However, when the nucleic acid exceeded a certain concentration, the enzyme activity began to decrease.

### 2.8. Synthesis of Tryptophan by Assembled Enzymes

Next, we used the wild-type enzyme and assembled enzyme to catalyze the synthesis of tryptophan under the optimal reaction conditions. The highest activity was observed for the assembled TrapB-dCE9 (Figure 17), which was about twice as active as wild-type TSase.

The above experiments proved that TrapB-dCE9 had the highest enzyme activity. We then used the labeled enzyme with exogenous nucleic acid to further improve the activity of TSase. After adding 0.4 pM of exogenous nucleic acid, the enzyme could produce 0.58 g/L of tryptophan under the optimal conditions, which is almost three times as much as the wild-type enzyme. This also confirmed that the introduction of dCE increased the enzyme’s activity due to its ability to bind endogenous nucleic acids. Compared with the activity of the enzyme without exogenous nucleic acid, the increase could be more than three-fold, which confirmed the feasibility of using nucleic acid scaffolds to enhance enzyme activity.

## 3. Materials and Methods

### 3.1. Strains and Plasmids

The main plasmids used in this experiment were pET23a and pET28a. *E. coli* XL10 Gold (Novagen, Madison, WI, USA) was used for cloning. *E. coli* BL21 (DE3) Condon was also obtained from Novagen (Madison, WI, USA).

### 3.2. Recombinant Plasmid Construction

The TSase gene (HAV8987736.1) was synthesized by Wuhan GeneCreate Biological Engineering Co., Ltd. (Wuhan, China). The dCE sequences (dCE2, dCE7, dCE8, dCE9) are shown in the Appendix A. The recombinant plasmids in this paper were also constructed by GeneCreate Biological Engineering Co. (Table 1).

### 3.3. Protein Expression and Purification

*E. coli* BL21 (DE3) was used as the host strain cell for protein expression. Bacteria were transformed with the plasmids, which were verified by sequencing, using routine protocols. After transformation, a single colony was inoculated in a liquid LB medium containing 100 μg/mL of kanamycin and incubated at 37 °C at 220 rpm shaking for 16–18 h. A small aliquot of this seed solution was added to the TB medium, which was further incubated at 37 °C at 220 rpm shaking. When the OD_600_ reached 0.6–0.8, IPTG (final concentration of 1 mM) was added to induce the expression of the recombinant protein, and the bacteria were incubated at 18 °C at 220 rpm shaking for ≥18 h. The cultures were centrifuged for 10 min at 6000× *g*. The supernatant was discarded, cells were resuspended in ultrapure water and centrifuged again, and the cell pellet was stored at −80 °C until use.

For protein analysis, PBS was added to the centrifuge tubes containing the bacterial cells. The cells were resuspended, and PMSF (final concentration 1 mM) was added. Then, a cell breaker was used to crush the cells through five or six repetitions at low temperature. The resulting liquid was divided into 2-mL EP tubes, which were centrifuged at 10,000× *g* for 20 min, and the supernatant was collected. The supernatant was poured into a packed, pretreated Ni-NTA purification column, which was placed on a silent mixer at 4 °C for 2 h. The supernatant was removed, and the column was washed with three volumes of PBS containing 10 mM, 30 mM, 100 mM, and 300 mM of imidazole (four washing steps in total). The flow-through at each step was collected. From each eluate, 80 μL was mixed with 20 μL 5× loading buffer, heated to 100 °C for 10 min, and centrifuged at 10,000× *g* for 1 min. After centrifugation, SDS-PAGE was performed to determine the purity of the target protein. The eluate with little protein but a high concentration of target protein was placed in an ultrafiltration tube for ultrafiltration concentration. After concentration, the solution was changed to PBS. After the fluid change, the samples were aliquoted, frozen in liquid nitrogen, and stored at −80 °C.

The gel filtration experiment was performed as follows: The dCE-TrapB and TrapA mixture passed through gel filtration using a Superdex 75 (GE healthcare, Chicago, IL, USA) in a 20 mM HEPES-KOH buffer (pH 7.9) supplemented with 200 mM of KCl and 1 mM of EDTA.

The procedures for DNA removal by heparin column were performed as follows: The cells were disrupted by sonication, and after centrifugation, a crude enzyme sample was prepared by heat treatment of the cytoplasmic fraction at 85 °C for 10 min. A heparin column (1.6 by 2.5 cm; HiTrap; Pharmacia, Hong Kong, China) was equilibrated with the buffer (10 mM of sodium phosphate [pH 7.0], 0.1 M of NaCl, 0.1 mM of EDTA, 1 mM of dithiothreitol, 10% glycerol), and the crude enzyme sample was applied to the column. Protein fractions were eluted by a linear gradient of 0.1 to 1.5 M of NaCl. The dCE-labeled TrapB was recovered at approximately 0.6 to 0.7 M.

A 12% (*w*/*v*) SDS-PAGE was used to detect TSase. The protein concentration was determined using the Bradford method.

### 3.4. Enzymatic Characterization of Tryptophan Synthetase

The amount of recombinant TSase required to generate 1 mol of L-tryptophan under certain reaction conditions was defined as 1 U. Tryptophan synthetase activity is calculated according to the amount of tryptophan produced.

### 3.5. Determination of L-Tryptophan Levels

For the determination of L-tryptophan content, two methods were selected. For qualitative detection, paper chromatography was employed. A horizontal line was drawn 1–2 cm from the lower end of the chromatographic plate, and several points on the horizontal line were marked according to the number of samples. Standards and samples were tested simultaneously. A development agent and color development agent were poured into the chromatography cylinder in a certain proportion, and the chromatography plate was placed inside (the line of the spot sample was higher than the chromatography page). When the layer was extended to 1–2 cm at the other end, the chromatography plate was taken out and placed in the oven for drying, so the spots could be observed.

HPLC was employed for quantitative analysis of L-tryptophan. An Agilent C18 separation column (5 μm, 4.6 mm × 250 mm) was used at a flow rate of 1 mL/min, a mobile phase of 0.03% KH_2_PO_4_:methanol (90:10), a wavelength of 278 nm, and an injection volume of 10 μL. All experiments were performed in triplicate. In order to assess the significance of the observed differences, a t-test was conducted.

## 4. Conclusions

We reported an exploratory experiment to analyze the effects of fusing modified colistin and TSase on tryptophan production. Modified CE did not exhibit DNase activity, but retained its ability to bind to nucleic acid. Therefore, we explored whether the introduction of a dCE protein as a nucleic acid scaffold could improve the activity of TSase. The output of the target product was indeed increased, providing a reference for the modification of similar enzymes, and the application of nucleic acid scaffolds in future studies.

In the early stages of this study, in order to verify whether the addition of dCE could improve the activity of TSase, and to explore differences in the effects of fusing the label to the two different subunits, the dCE label was first fused to the C-terminus of the α subunit. The introduction of the tag indeed enhanced the activity of TSase, but the effect was not so obvious. The label was then fused to the N-terminus of the β subunit, and it was found that the catalytic activity of the cells improved greatly. However, during the purification process, after fusion to the N-terminus, the label might break, leading to a change in the ratio between the two subunits. Therefore, the two subunits were expressed separately, and the proteins were combined in vitro to ensure the equimolar combination between the two subunits. The carrier was rebuilt, and the label was fused to the C-terminus of the β subunit, and the catalytic activity of whole cells and the purified enzyme was determined. The introduction of a dCE tag improved TSase activity.

## Figures and Tables

**Figure 1 molecules-28-07272-f001:**
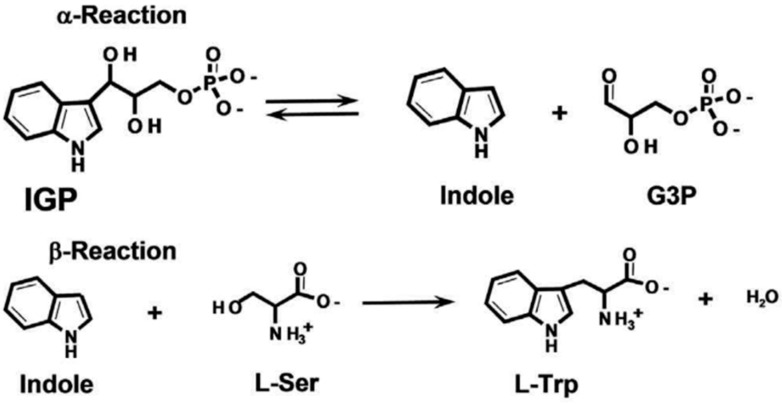
Tryptophan synthase reaction mechanism.

**Figure 2 molecules-28-07272-f002:**
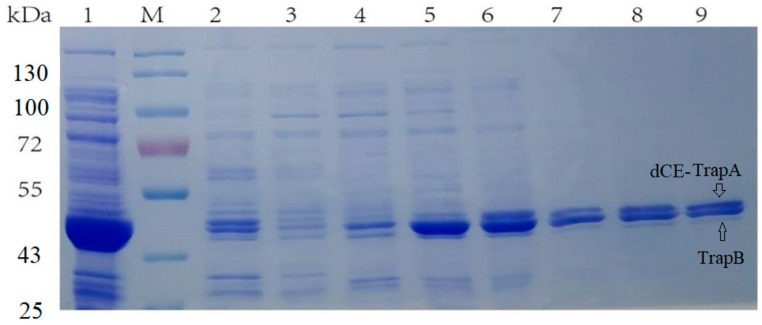
SDS-PAGE analysis of the Ectrp-dCE2 expression. M: Protein Marker. Lane 1: Breaking supernatant of Ectrp-dCE2, Lane 2: flow-through of Ectrp-dCE2; Lane 3: 10 mM imidazole eluent; Lane 4: 30 mM imidazole eluent; Lane 5–6: 100 mM imidazole eluent; Lane 7–9: 300 mM imidazole eluent.

**Figure 3 molecules-28-07272-f003:**
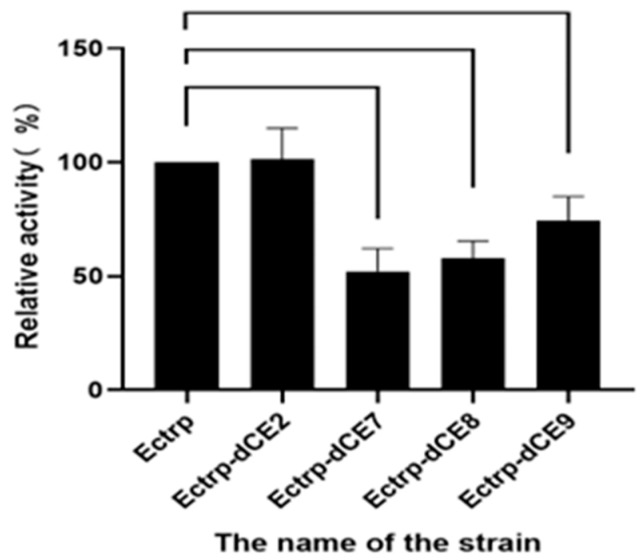
The synthesis of L-tryptophan using different dCE-labeled recombinant strains (Ectrp-dCE2, Ectrp-dCE7, Ectrp-dCE8, and Ectrp-dCE9).

**Figure 4 molecules-28-07272-f004:**
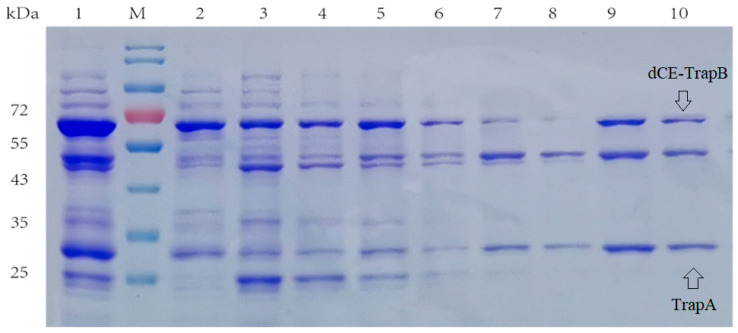
SDS-PAGE analysis of the dCE2-Ectrp expression. M: Protein Marker. Lane 1: Breaking supernatant of dCE2-Ectrp, Lane 2: flow-through of dCE2-Ectrp; Lane 3: 10 mM imidazole eluent; Lane 4: 30 mM imidazole eluent; Lane 5–7: 100 mM imidazole eluent; Lane 8–10: 300 mM imidazole eluent.

**Figure 5 molecules-28-07272-f005:**
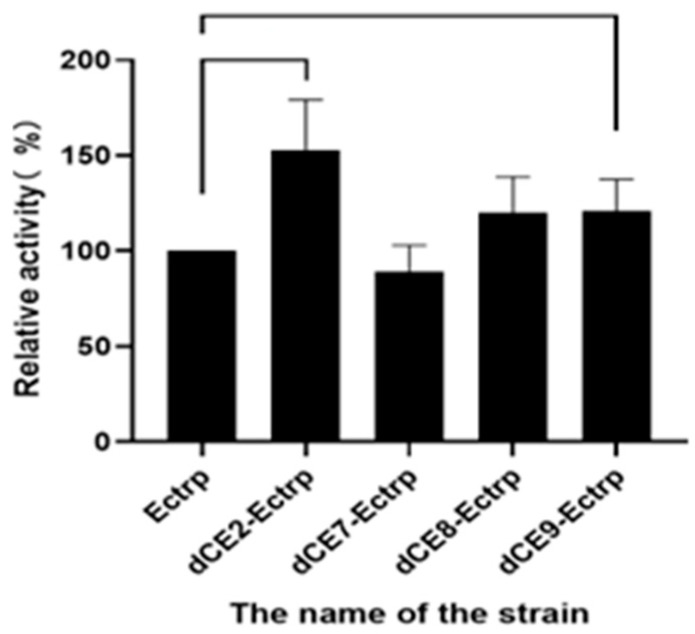
The synthesis of L-tryptophan using different dCE-labeled recombinant strains (dCE2-Ectrp, dCE7-Ectrp, dCE8-Ectrp, and dCE9-Ectrp).

**Figure 6 molecules-28-07272-f006:**
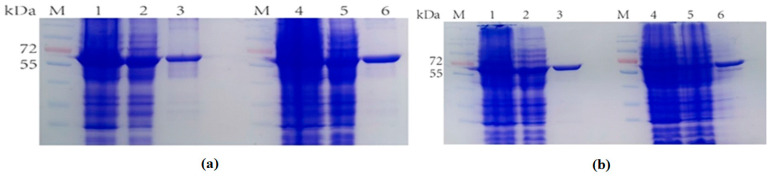
(**a**) M: Protein Marker. Lane 1: Breaking supernatant of TrapB-dCE2; Lane 2: flow-through of TrapB-dCE2; Lane 3: purified TrapB-dCE2; Lane 4: Breaking supernatant of TrapB-dCE7; Lane 5: flow-through of TrapB-dCE7; Lane 6: purified TrapB-dCE7; (**b**) M: Protein Marker. Lane 1: Breaking supernatant of TrapB-dCE8; Lane 2: flow-through of TrapB-dCE8; Lane 3: purified TrapB-dCE8; Lane 4: Breaking supernatant of TrapB-dCE9; Lane 5: flow-through of TrapB-dCE9; Lane 6: purified TrapB-dCE9.

**Figure 7 molecules-28-07272-f007:**
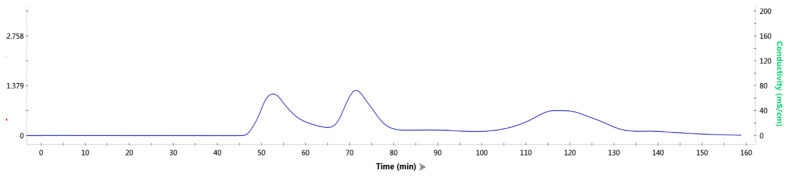
Gel filtration results of mixed protein liquid.

**Figure 8 molecules-28-07272-f008:**
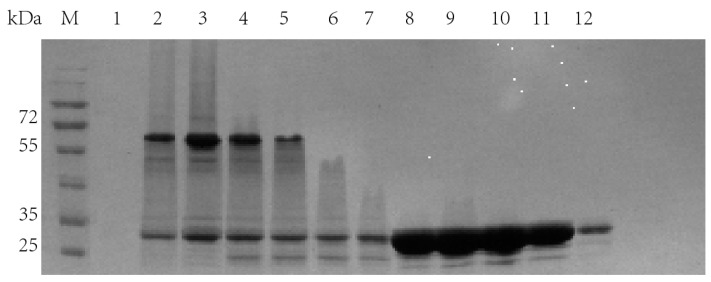
SDS-PAGE results of molecular sieve chromatography single peak electrophoresis. Lanes 1–6: the sampling results corresponding to the first peak in the molecular sieve chromatography; Lane 7–12: sampling results corresponding to the second peak in the molecular sieve chromatography.

**Figure 9 molecules-28-07272-f009:**
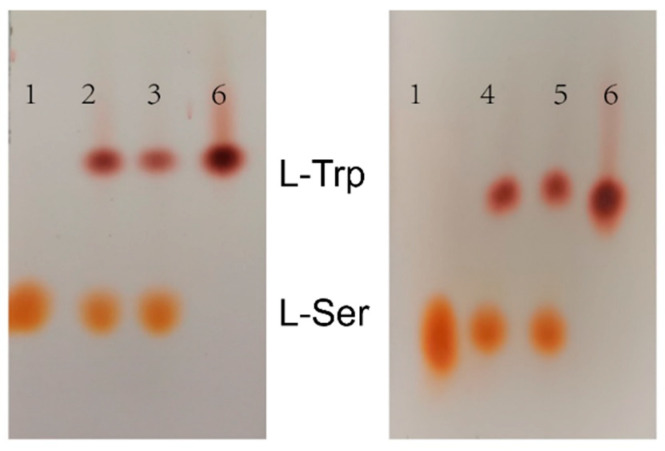
TLC results of assembled tryptophan synthase reaction. 1: 1% L-Ser standard; 2: Reaction solution of TrapB-dCE2/TrapA; 3: Reaction solution of TrapB-dCE7/TrapA; 4: Reaction solution of TrapB-dCE8/TrapA; 5: Reaction solution of TrapB-dCE9/TrapA; 6: 1% of L-Trp standard.

**Figure 10 molecules-28-07272-f010:**
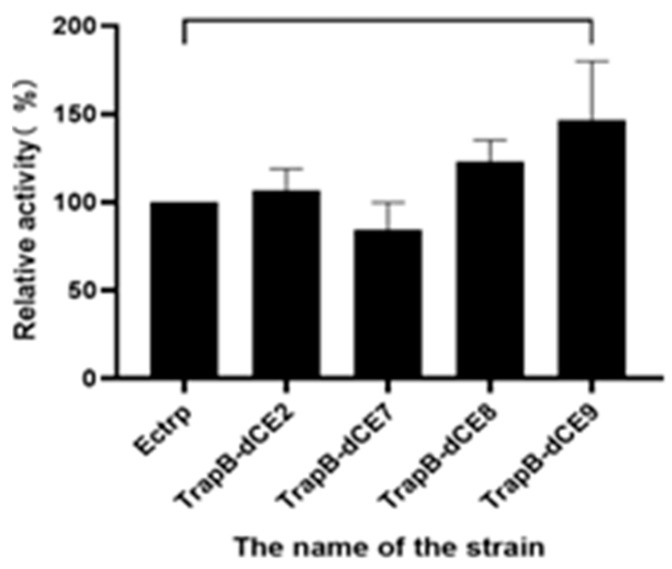
HPLC results of assembled tryptophan synthase reaction.

**Figure 11 molecules-28-07272-f011:**
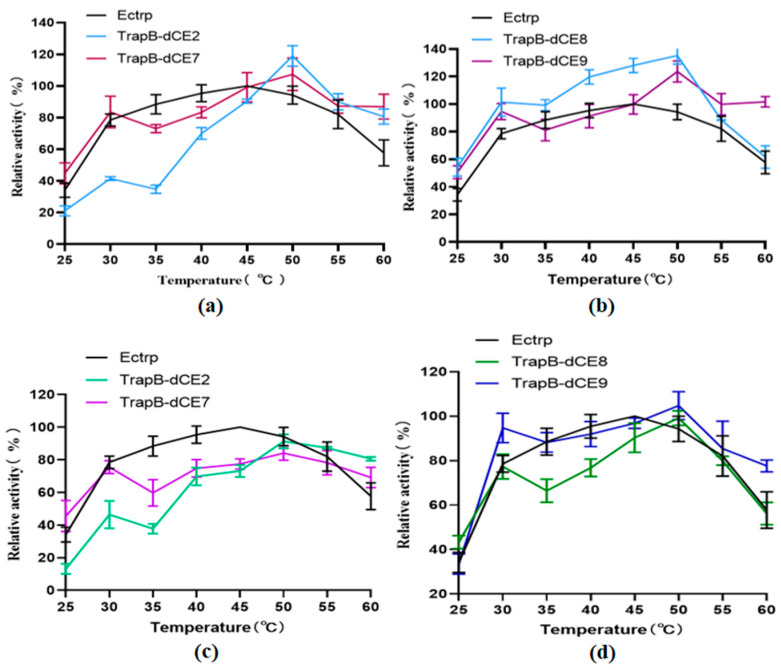
(**a**): The relative activity of assembled TrapB-dCE2/TrapA, TrapB-dCE7/TrapA, and Ectrp at different temperatures; (**b**): The relative activity of assembled TrapB-dCE8/TrapA, TrapB-dCE9/TrapA, and Ectrp at different temperatures; (**c**) After nucleic acid removal, the relative activity of TrapB-dCE2/TrapA, TrapB-dCE7/TrapA with Ectrp at different temperatures; (**d**) After nucleic acid removal, the relative activity of trapB-dCE8, trapB-dCE9, and Ectrp at different temperatures. All these experiments were performed at pH 9.0.

**Figure 12 molecules-28-07272-f012:**
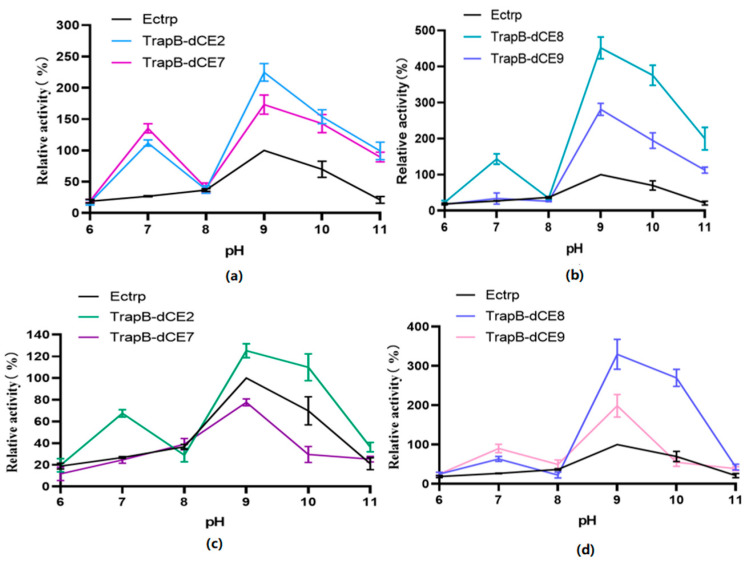
(**a**): The relative activity of assembled TrapB-dCE2/TrapA, TrapB-dCE7/TrapA, and Ectrp at different pH; (**b**): The relative activity of assembled TrapB-dCE8/TrapA, TrapB-dCE9/TrapA, and Ectrp at different pH; (**c**) After nucleic acid removal, the relative activity of TrapB-dCE2/TrapA, TrapB-dCE7/TrapA with Ectrp at different pH; (**d**) After nucleic acid removal, the relative activity of trapB-dCE8, trapB-dCE9, and Ectrp at different pH. All these experiments were performed at 50 °C.

**Figure 13 molecules-28-07272-f013:**
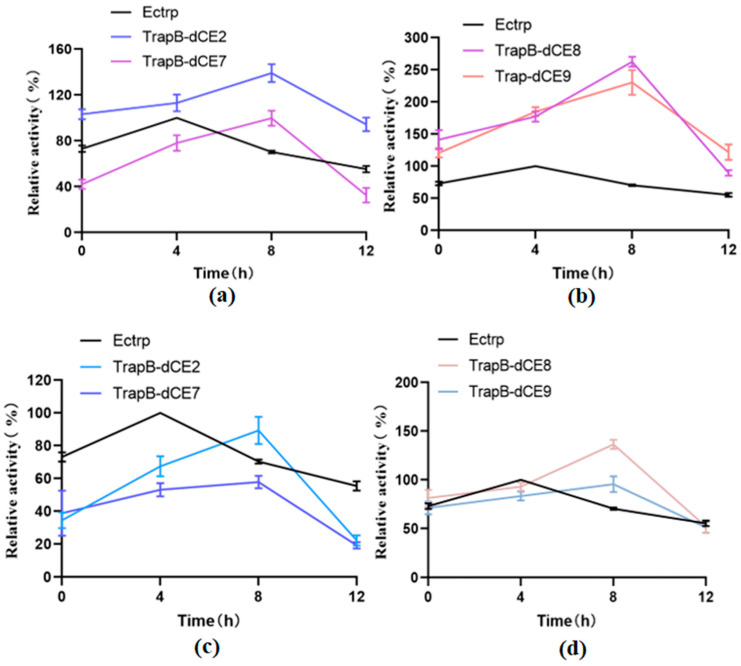
(**a**): The relative activity of assembled TrapB-dCE2/TrapA, TrapB-dCE7/TrapA, and Ectrp at different times; (**b**): The relative activity of assembled TrapB-dCE8/TrapA, TrapB-dCE9/TrapA, and Ectrp at different times; (**c**) After nucleic acid removal, the relative activity of TrapB-dCE2/TrapA, TrapB-dCE7/TrapA with Ectrp at different times; (**d**) After nucleic acid removal, the relative activity of TrapB-dCE8/TrapA, TrapB-dCE9/TrapA, and Ectrp at different times. All these reactions were performed at pH 9.0 and 50 °C.

**Figure 14 molecules-28-07272-f014:**
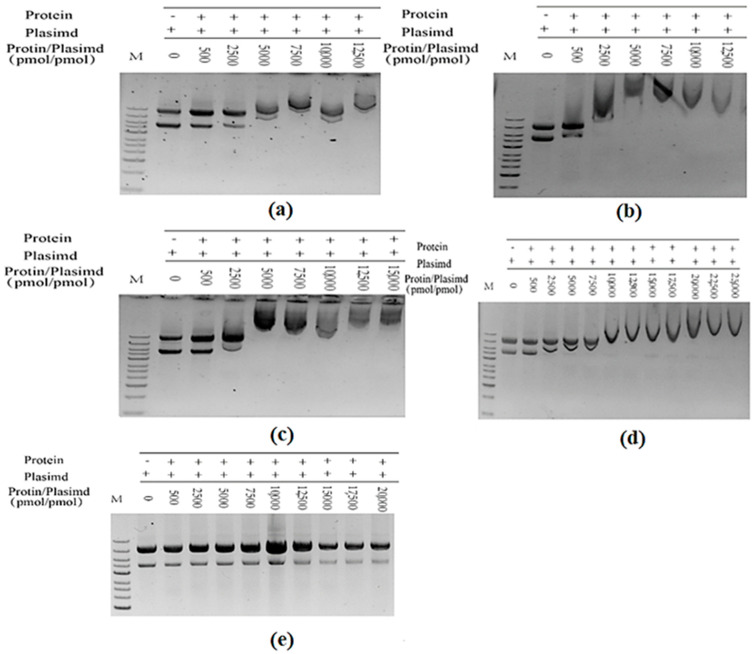
(**a**) The purified TrapB-dCE2 (removal of endogenous nucleic acids) combined with exogenous nucleic acid; (**b**) The purified TrapB-dCE7 (removal of endogenous nucleic acids) combined with exogenous nucleic acid; (**c**) The purified TrapB-dCE8 (removal of endogenous nucleic acids) combined with exogenous nucleic acid; (**d**) The purified TrapB-dCE9 (removal of endogenous nucleic acids) combined with exogenous nucleic acid; (**e**) Wild-type combined with exogenous nucleic acid.

**Figure 15 molecules-28-07272-f015:**
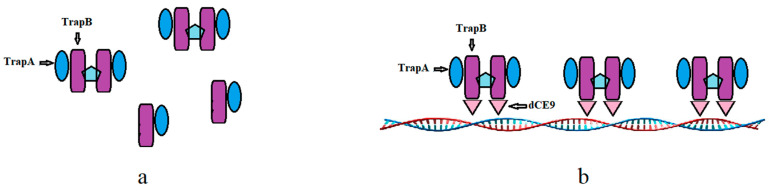
The wild TSase (**a**) and assembled TSase (**b**). The dCE9, TrapA, and TrapB sequences can be found in the Appendix A.

**Figure 16 molecules-28-07272-f016:**
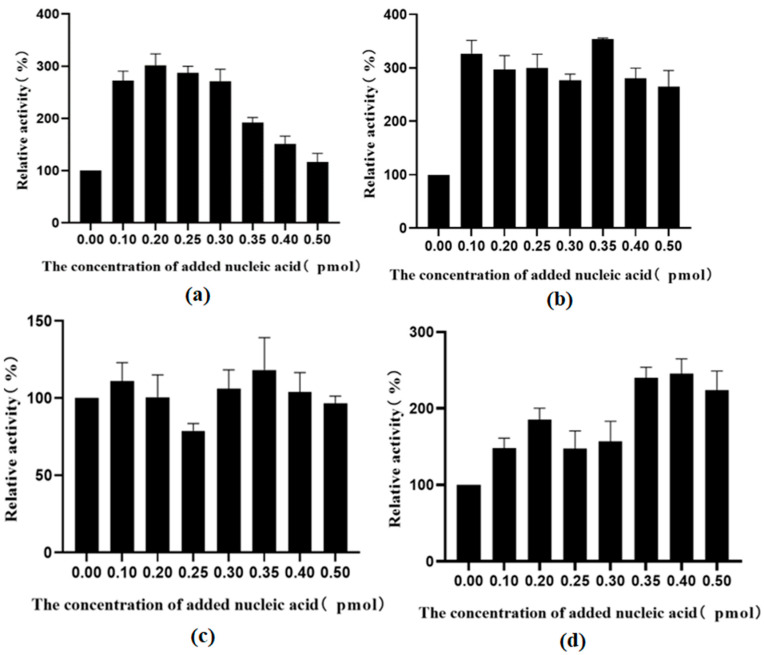
(**a**) The effect of different concentrations of nucleic acid on the activity of TrapB-dCE2 (removal of endogenous nucleic acids); (**b**) The effect of different concentrations of nucleic acid on the activity of TrapB-dCE7 (removal of endogenous nucleic acids); (**c**) The effect of different concentrations of nucleic acid on the activity of TrapB-dCE8 (removal of endogenous nucleic acids); (**d**) The effect of different concentrations of nucleic acid on the activity of TrapB-dCE9 (removal of endogenous nucleic acids).

**Figure 17 molecules-28-07272-f017:**
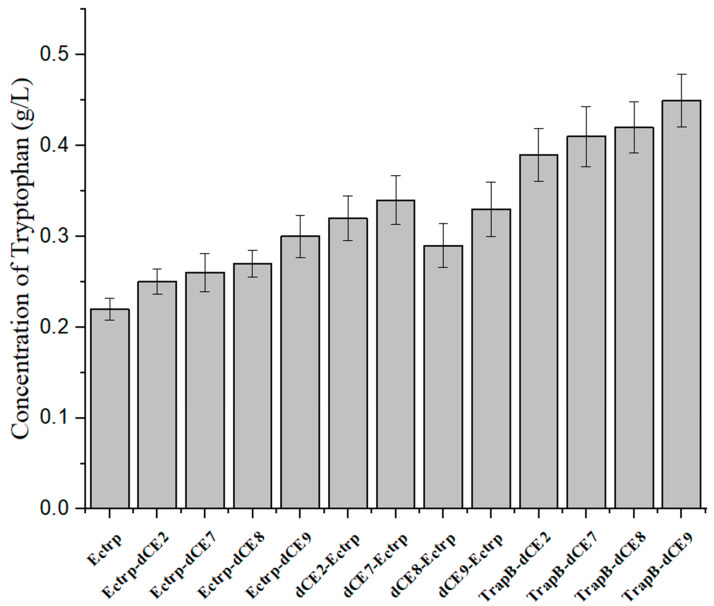
The production of tryptophan using labeled enzyme with exogenous nucleic acid.

**Table 1 molecules-28-07272-t001:** Strains and plasmids used in this study.

Plasmids	Features	Source
pET28a-Ectrp	Promoter T7, PBR322 Ori, *Kan^r^*, N-His	Our lab
pET23a-dCE2-	Promoter T7, PBR322 Ori, A*mp^r^*	Our lab
pET23a-dCE7	Promoter T7, PBR322 Ori, A*mp^r^*	Our lab
pET23a-dCE8	Promoter T7, PBR322 Ori, A*mp^r^*	Our lab
pET23a-dCE9	Promoter T7, PBR322 Ori, A*mp^r^*	Our lab
pET23a-Ectrp	Promoter T7, PBR322 Ori, A*mp^r^*, N-His, C-His	Our lab
pET23a-Ectrp-dCE2	Promoter T7, PBR322 Ori, A*mp^r^*, N-His, C-His	This paper
pET23a-Ectrp-dCE7	Promoter T7, PBR322 Ori, A*mp^r^*, N-His, C-His	This paper
pET23a-Ectrp-dCE8	Promoter T7, PBR322 Ori, A*mp^r^*, N-His, C-His	This paper
pET23a-Ectrp-dCE9	Promoter T7, PBR322 Ori, A*mp^r^*, N-His, C-His	This paper
pET23a-dCE2-Ectrp	Promoter T7, PBR322 Ori, A*mp^r^*, N-His, C-His	This paper
pET23a-dCE7-Ectrp	Promoter T7, PBR322 Ori, A*mp^r^*, N-His, C-His	This paper
pET23a-dCE8-Ectrp	Promoter T7, PBR322 Ori, A*mp^r^*, N-His, C-His	This paper
pET23a-dCE9-Ectrp	Promoter T7, PBR322 Ori, A*mp^r^*, N-His, C-His	This paper
pET23a-TrapA	Promoter T7, PBR322 Ori, A*mp^r^*, C-His	This paper
pET23a-TrapB-dCE2	Promoter T7, PBR322 Ori, A*mp^r^*, N-His	This paper
pET23a-TrapB-dCE7	Promoter T7, PBR322 Ori, A*mp^r^*, N-His	This paper
pET23a-TrapB-dCE8	Promoter T7, PBR322 Ori, A*mp^r^*, N-His	This paper
pET23a-TrapB-dCE9	Promoter T7, PBR322 Ori, A*mp^r^*, N-His	This paper
pET23a-dCE2-TrapB	Promoter T7, PBR322 Ori, A*mp^r^*, C-His	This paper
pET23a-dCE7-TrapB	Promoter T7, PBR322 Ori, A*mp^r^*, C-His	This paper
pET23a-dCE8-TrapB	Promoter T7, PBR322 Ori, A*mp^r^*, C-His	This paper
pET23a-dCE9-TrapB	Promoter T7, PBR322 Ori, A*mp^r^*, C-His	This paper

## Data Availability

Not applicable.

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
