# Peer review of "Improving the Activity of Tryptophan Synthetase via a Nucleic Acid Scaffold"

_molecules, 2023, doi:10.3390/molecules28217272_

Round 1

Reviewer 1 Report

Wang et al. added a nucleic acid scaold to tryptophan synthetase to improve its catalytic activity. The experiments results indicated that by adding of nucleic acid tag and fusion of subunits, the enzymic activity was enhanced largely. The approaches and test results could be valuable in production of amino acid and enzyme-modification. However, authors did need to pay more time and attention to data organization. e.g., the quality of figures and tables should be improved, which were in low resolution and poor presentation. There were no structure or scheme figure to show the position of nucleic acid scaold or architecture of modified enzyme complex, which is quite bad for readers to understand the results and idea of authors’ work. Overview, this is a research article, author should pay attention to scientific writing and enough discussion for readers to catch the highlights of this work at least easily and quickly.

Author Response

Dear reviewer:

Respond:

In the revised paper, we have added the scheme figure to show nucleic acid scaffold. And made major revision in many aspects. These were shown in red.

Reviewer 2 Report

The manuscript entitled "Improving the activity of tryptophan synthetase by a nucleic acid scaffold" provides interesting and worthy information.

In my opinion, some points should be corrected/improved.

General comments:

-English language should be improved.

-It should be indicated  that the TrapA protein is α unit, whereas TrapB protein is β unit of Tryptophan synthetase .

Also, what about Trp B (page 5, row 187)?

-A list of abbreviation will be useful for readers.

Section Results and discussion:

Figure 1 (legend) and Figure 3. (legend):

-An explanation (caption) of the results presented should be added. For example: SDS-PAGE analysis of...

-10mm imidazole eluent.... 300 mm imidazole eluent"; This should be corrected: 10mM imidazole eluent...

Figure 1: Why only two masses ( 72 and 55 kDa) are labeled?

Page 4, row 164, It is written:

" The size of the TrapA protein with label is about  44 kDa, and the size of the TrapB protein without label was about 47 kDa. Both bands could be clearly seen in the figure."

My suggestion is to mark (indicate) these bands with arrows in the figure.

Figures 2, 4, and 9:

-It is obvious that some Statistical Tests were used for the data analysis. These Statistical Tests should be specified in the text. Accordingly, the presented results should be explained.

Figure 10. The relative activity of assembled ... at different temperatures.

What pH was used for the investigation of relative activity dependence on temperature?

Figure 11. The relative activity of assembled .... at different pH.

What temperature was used for the investigation of relative activity dependence on pH?

Section, 3.12. Synthesis of tryptophan by assembled enzymes

A figure caption is missing.

Author Response

The manuscript entitled "Improving the activity of tryptophan synthetase by a nucleic acid scaffold" provides interesting and worthy information.

In my opinion, some points should be corrected/improved.

General comments:

-English language should be improved.

Respond:

In the revised paper, we have improved English language.

-It should be indicated  that the TrapA protein is α unit, whereas TrapB protein is β unit of Tryptophan synthetase .

Respond:

In the revised paper, we have indicated TrapA protein is α unit, whereas TrapB protein is β unit.

Also, what about Trp B (page 5, row 187)?

-A list of abbreviation will be useful for readers.

 Respond:

Sorry, Trp B is wrong, it should be TrapB.

Section Results and discussion:

Figure 1 (legend) and Figure 3. (legend):

-An explanation (caption) of the results presented should be added. For example: SDS-PAGE analysis of...

Respond:

The legend has been revised in the resubmitted paper.

-10mm imidazole eluent.... 300 mm imidazole eluent"; This should be corrected: 10mM imidazole eluent...

Respond:

This was revised in the paper.

Figure 1: Why only two masses ( 72 and 55 kDa) are labeled?

Page 4, row 164, It is written:

" The size of the TrapA protein with label is about  44 kDa, and the size of the TrapB protein without label was about 47 kDa. Both bands could be clearly seen in the figure."

My suggestion is to mark (indicate) these bands with arrows in the figure.

Respond:

We have revised these in the paper as suggested.

Figures 2, 4, and 9:

-It is obvious that some Statistical Tests were used for the data analysis. These Statistical Tests should be specified in the text. Accordingly, the presented results should be explained.

Respond:

This was revised in the paper.

Figure 10. The relative activity of assembled ... at different temperatures.

What pH was used for the investigation of relative activity dependence on temperature?

 Respond:

Temperature was added in the paper.

Figure 11. The relative activity of assembled .... at different pH.

 Respond:

pH was added in the paper.

What temperature was used for the investigation of relative activity dependence on pH?

Respond:

Temperature was added in the paper.

Section, 3.12. Synthesis of tryptophan by assembled enzymes

A figure caption is missing.

Respond:

The figure caption was added in the paper.

Reviewer 3 Report

The authors present a work in which Tryptophan synthetase (TSase) was fused with a modified Colistin protein (dCE) that can bind DNA in order to enhance enzyme activity and stability. This idea is based on the fact that in cells proteins exist in a quite crowded environment in which also DNA can form scaffolds that stabilize and control enzyme activity. The aim is clear and the work could be of interest in the field of enzymology and biocatalysis. However, the manuscript must be improved because many methodological aspects are not clear (see issues below). Moreover, authors must reread carefully the text to check spelling errors (also in figures) and improve the description of the work.

Issues to be addressed:

- in the abstract, CE must be explicited.

- it would be good to show a figure with the reaction of TSase.

- line 58: what do the authors mean with "fifth channel"?

- line 96-97: what do authors mean "dCE sequences will be published in another paper"? All data regarding the constructs that are used must be explicited. How are the two subunits combined in the fusion constructs? A figure showing the constructs should be included.

- Table 1: why do some constructs have His-tag at both N and C termini?

- line 119: which protein gravity purification column?

- line 163-167 and 187-192: the need for a better description of the constructs mentioned above arises also here. When cells are transformed with TrapA construct, how it could be that also TrapB is found? And viceversa in the case of cells transformed with TrapB construct. Authors must clarify.

- line 233: what do authors mean with molecular sieve chromatography? I guess it's a gelfiltration... This is neither mentioned in the methods section. What is the elution buffer? Also buffer conditions and procedures for DNA removel by heparin column mentioned in line 272 are not properly described.

- Figs. 10 and 11: what assay was used to test activity?

- line 345: it should be specified that it's a DNA mobility shift assay.

- in the comparison of relative activity of the various constructs it is not clear what is the reference. What is the activity of TSase without CE? This is mentioned in section 3.12 but it is not clear the comparison among wild type, assembled enzyme, fusion enzyme with CE etc.

- Figure in page 14 does not have number and legend.

Apart from some spelling errors mentioned above, it is not English itself to be improved but more the style used to describe the various concepts and the steps.

Author Response

The authors present a work in which Tryptophan synthetase (TSase) was fused with a modified Colistin protein (dCE) that can bind DNA in order to enhance enzyme activity and stability. This idea is based on the fact that in cells proteins exist in a quite crowded environment in which also DNA can form scaffolds that stabilize and control enzyme activity. The aim is clear and the work could be of interest in the field of enzymology and biocatalysis. However, the manuscript must be improved because many methodological aspects are not clear (see issues below). Moreover, authors must reread carefully the text to check spelling errors (also in figures) and improve the description of the work.

Issues to be addressed:

- in the abstract, CE must be explicited.

Respond:

CE was explained in the revised paper.

- it would be good to show a figure with the reaction of TSase.

Respond:

Figure 1 showed the reaction of TSase.

- line 58: what do the authors mean with "fifth channel"?

Respond:

It was deleted.

- line 96-97: what do authors mean "dCE sequences will be published in another paper"? All data regarding the constructs that are used must be explicited. How are the two subunits combined in the fusion constructs? A figure showing the constructs should be included.

Respond:

dCE sequences were added in the revised paper. Figure. 15 was added and showed the constructs.

- Table 1: why do some constructs have His-tag at both N and C termini?

Respond:

To make purification easier.

- line 119: which protein gravity purification column?

Respond:

It was Ni column and revised in the paper.

- line 163-167 and 187-192: the need for a better description of the constructs mentioned above arises also here. When cells are transformed with TrapA construct, how it could be that also TrapB is found? And viceversa in the case of cells transformed with TrapB construct. Authors must clarify.

Respond:

The gene fragment of TSase is also composed of the TrapA gene and the TrapB gene placed in tandem. So they were expressed in one reading frame.

- line 233: what do authors mean with molecular sieve chromatography? I guess it's a gelfiltration... This is neither mentioned in the methods section. What is the elution buffer? Also buffer conditions and procedures for DNA removel by heparin column mentioned in line 272 are not properly described.

Respond:

These were added in the methods section.

- Figs. 10 and 11: what assay was used to test activity?

Respond:

HPLC was used to test activity.

- line 345: it should be specified that it's a DNA mobility shift assay.

Respond:

DNA mobility shift assay was added in the paper.

- in the comparison of relative activity of the various constructs it is not clear what is the reference. What is the activity of TSase without CE? This is mentioned in section 3.12 but it is not clear the comparison among wild type, assembled enzyme, fusion enzyme with CE etc.

Respond:

These were clear in the paper. The activity of TSase is as the reference.

- Figure in page 14 does not have number and legend.

Respond:

The legend was added in the revised paper.

Round 2

Reviewer 3 Report

Most of isuues were addressed, but there are still some that need attention:

- figure 1 is of very low graphical quality

- in abstract CE is still not explicited

- lines 101-102 are still the same (containing also a typo error): "The dCE sequences (dCE2, dCE7, dCE8, 101, dCE9) will be published in anther paper." Figure 15 is a schematic representation of the constructs, but the authors should specify which amino acid range in contained in each construct.

Author Response

Most of isuues were addressed, but there are still some that need attention:

- figure 1 is of very low graphical quality

Respond:

A new figure was supplied in the revised paper.

- in abstract CE is still not explicited

Respond:

It was explicated in revised paper.

- lines 101-102 are still the same (containing also a typo error): "The dCE sequences (dCE2, dCE7, dCE8, 101, dCE9) will be published in anther paper." Figure 15 is a schematic representation of the constructs, but the authors should specify which amino acid range in contained in each construct.

Respond:

The error has been corrected. The amino acid range was added in the revised paper.
